# Delivering Integrated Community Care for the Elderly: A Qualitative Case Study in Southern China

**DOI:** 10.3390/ijerph21060680

**Published:** 2024-05-25

**Authors:** Yiqiao Li, Lan Luo, Hongying Dong

**Affiliations:** 1School of Marxism, South China Normal University, Guangzhou 510631, China; 20230006@m.scnu.edu.cn; 2Huangpu Hongshanjie Community Health Service Center, Guangzhou 510725, China

**Keywords:** integrated community care, aging in place, healthy ageing, elderly care service

## Abstract

The rapid aging and increasing care demands among the elderly population present challenges to China’s health and social care system. The concept of aging in place has prompted the implementation of integrated community care (ICC) in the country. This study aims to provide empirical insights into the practices of integrated care policies and approaches at the community level. Data for this study were collected through six months of participatory observations at a local community health service center in a southern Chinese city. Semi-structured interviews were conducted with the multidisciplinary community care team to gather frontline formal caregiver perceptions of ICC, thereby facilitating a better understanding of the obstacles and opportunities. Qualitative analysis revealed four themes: the ICC delivery model and development strategies within the community care scheme, the person-centered guiding principle, and the challenges and struggles encountered by formal caregivers within China’s current ICC system. The case study presented herein serves as a notable example of the pivotal role of primary care in the successful implementation of elderly care within a community setting. The adoption of a private organization-led approach to medico-social integration care in the community holds significant potential as a service delivery model for effectively addressing a wide range of elderly care issues.

## 1. Introduction

Population aging presents significant challenges to both society and public health, particularly in developing regions such as China, which is experiencing the world’s fastest-growing aging population. As older adults age in their own homes, they often require physical, health, and/or social care assistance due to physical and cognitive decline. Consequently, there has been a surge in demand for elder care services in the country, exacerbating the shortage of public health services, institutional care facilities, integrated medical and care systems, long-term care insurance programs, and supportive environments [1].

Adapting to the escalating care needs of the older population poses formidable challenges for China’s health and social care systems. To address these challenges, the country has embraced the concept of aging in place and incorporated it into its long-term elder care strategy since the mid to late 2000s. In 2015, the Chinese central government proposed to build a multi-level elderly care service system with home-based care as the foundation, community as support, and institutions as supplements [2]. In Report on the Work of the Government 2016, it was put forward to launch comprehensive reform pilots in the elderly care service industry [3]. In response to the deployment of the central government, during the “13th Five-Year Plan” period (from 2016 to 2020), the Ministry of Civil Affairs and the Ministry of Finance initiated reform nationwide, including 31 provinces and districts, where it carried out home-based and community-based elderly care service reform pilots. After seven years of pilot experimentation, a total of 203 pilot areas were established throughout the country. With the implementation of pilot reforms in home-based and community elderly care services, the Chinese government has gradually improved the formulation of relevant policies [1]. In 2021, the CPC Central Committee and the State Council issued the “Opinions on Strengthening Elderly Work in the New Era” and the “14th Five-Year Plan for National Aging Development and Elderly Care Service System,” proposing to prioritize home-based elderly care and strengthen the development goal of home and community elderly care service capabilities [4].

In 2023, the General Office of the CPC Central Committee and the General Office of the State Council issued the “Opinions on Promoting the Construction of Basic Elderly Care Service System”. A major highlight of this document is the “National Basic Elderly Care Service List”. In this list, a guarantee of the elderly population’s basic living security is the mainline. The overall service items in existing effective laws, regulations, and policy documents were systematically summarized and integrated into the institutional framework of basic elderly care services. The list includes 16 service items in three major categories: physical and financial assistance, care services, and social care services. At the same time, nearly 30 provincial governments were formulated, and follow-up implementation plans and elderly care service lists were published in their respective regions. Providing basic elderly care services to all elderly people in a standardized and categorized manner as a public product is an innovative policy measure [5]. 

In addition to improving the institutional policies of community-based and home-based elderly care services, the Chinese government also gradually upgraded the standardization of elderly care services. In July 2023, the implementation guide for the “Classification and Evaluation of Elderly Care Institutions” national standard was released [6]. In October 2023, the national standard “Basic Specification for At-home Care Services of the Elderly” (GB/T 43153-2023) was issued, which was the first national standard for home-based elderly care services in China [7]. In November 2023, the Ministry of Civil Affairs issued three recommended industry standards for elderly care services, including Specification for Rehabilitation Service of Senior Care Organization (MZ/T 205—2023) [8], Specification for Home-based Rehabilitation Service of the Elderly (MZ/T 206—2023) [9], and Specification for Bath Assistance Service of the Elderly (MZ/T 207—2023) [10]. In December, the Guidelines for Dementia Friendly Environments in Elderly Care Facilities, Guidelines for Aging Dwelling Environment Suitability Assessment and five industry standards were issued to solicit public opinion [11]. In addition, the National Health Commission and three other departments issued the “Guidelines for Home and Community Medical-nursing Combined Services (Trial)”, in which the need to provide health education, health management, medical rounds, home hospital beds, and other services for elderly people who need medical-nursing-combined services at home or in the community were stressed [12]. These governmental guidelines will provide ground rules for defining the scope of home-based elderly care services, regulating the service provider qualifications and specifying the service process contents, which is of practical significance for promoting the systematized and professional development of home-based elderly care services [13].

By of the end of 2021, the number of elderly individuals aged 65 and above receiving health management services in urban and rural communities nationwide reached 119 million. There were 4685 secondary and tertiary hospitals with geriatric departments, along with 15,431 primary healthcare centers as elderly friendly medical institutions, established. Additionally, 1027 medical and healthcare institutions providing palliative care services were also established [14]. By the end of 2022, the total number of healthcare personnel reached 812 million, including 94 million practicing physicians and 105 million registered nurses. The ratio of practicing physicians per thousand people was 3.15, and the ratio of registered nurses per thousand people was 3.71. Additionally, there were 3.28 general practitioners per ten thousand people and 6.94 personnel from specialized public health institutions per ten thousand people [15]. 

Community health institutions play a pivotal role in the primary healthcare system, delivering comprehensive and ongoing health services to community members. These services encompass various facets of medical care, public health, rehabilitation, immunization, health counseling, etc., ensuring integrated support for individuals’ health needs [16]. Among all the medical institutions nationwide, there were 10,353 community health service centers with 588,000 medical and nursing staff, averaging 57 personnel per center [15]. Community health centers typically operate as welfare-oriented and non-profit organizations, with most institutions established by the government. Community health workers (CHWs) consist of two groups: permanent staff directly employed by the government and temporary workers engaged through contracts by community health institutions [17].

In a community-based setting, care for older individuals is provided by integrating existing human resources, allowing them to receive day care services either through home visits or remotely at daycare centers while remaining in their own homes [18]. Essentially, integrated community care enables older adults to maintain their independence at home while receiving appropriate care and support from external sources, such as in-home or community care services [19].

However, ensuring adequate care for elderly individuals with prolonged needs in the most compassionate and least restrictive environment remains a significant social challenge [20]. Within the community setting, the quality and accessibility of healthcare and social care services profoundly influence the well-being of older people. A well-integrated system of primary and community care plays a crucial role in providing these services, with primary care serving as the cornerstone for addressing the complex needs of the elderly [21]. China has embarked on an innovative path aimed at improving access to integrated care for older individuals. This initiative involves establishing a community-based social and healthcare system that not only addresses chronic disease management and prevention, but also strengthens overall health services [22]. Furthermore, this system is supported by a robust healthcare workforce, facilitating its effective implementation and long-term sustainability. Although the Chinese government has encouraged multi-sector cooperation (including public–private and third sector partnerships) to deliver community eldercare services, a nationally standardized community eldercare system has yet to fully materialize [18]. Nevertheless, older individuals in China continue to face significant unmet care needs due to a shortage of informal caregiving resources and insufficient formal care, whether in institutional or residential settings. The challenge of implementing integrated community care is particularly evident in countries where healthcare and social services are overseen by separate government departments, with healthcare further divided into distinct departments for preventive, primary, secondary, and tertiary care within hospitals. Effectively navigating these rigid boundaries is essential for successfully implementing integrated care models within such systems.

Despite the crucial role of integrated community care for older individuals, its functions and significance remain inadequately explored. Most studies have been conducted at a strategic macro or national level, focusing on health system and policy implementation. For instance, they have examined integrated care policy implementation [23,24,25], primary care, delivery systems, and care coordination at the organizational level [26,27]. However, as highlighted in a review study, the majority of research focuses on coordinating care at the clinical (micro) level and service/organizational (meso) level, with limited exploration of system (macro) level integration and multilevel integrated care approaches [28]. Presently, there is a lack of clarity regarding the practical implementation of integrated care policies and approaches within communities, particularly from the perspective of frontline formal caregivers. Guided by comprehensive geriatric assessment [29], this study initiates by offering a descriptive account of a localized integrated community care model in China, demonstrating how interdisciplinary collaboration is fostered across various levels. Subsequently, this study delves into the perspectives of formal caregivers within the case study, seeking to gain profound insights into the opportunities and challenges encountered during the implementation of integrated community care practices.

## 2. Research Methods

### 2.1. Case Selection

The data for this research were collected over a six-month period of fieldwork conducted in Guangzhou, a southern city and the capital of Guangdong Province, China. As one of China’s most developed megacities, Guangzhou boasts a population of 18.73 million, with 1.95 million individuals aged 60 and above, constituting 18.86% of the total population. Beginning in June 2023, access was gained to the HS Community Health Center (the HS CHC), which is located in the HS Street Subdistrict of Huangpu District in the eastern part of the city. (The location and other such information of the case study was anonymized, consistent with the Chinese Pinyin abbreviations.) Within this locality, there are 4118 residents aged 60 and above, accounting for 22.5% of the overall population. The neighborhood is transitioning into a moderate aging stage, with an anticipated increase in the demand for long-term services and support, as well as a rise in the prevalence of disability and functional limitations. These factors make the community an ideal setting for observing the dynamics of aging in place. Based on its robust medical resources and comprehensive health services encompassing prevention, promotion, curative, rehabilitative, palliative, and end-of-life care, the HS CHC formally launched the Combining of Medical and Health Services and Elderly Care Service program in February 2021. This program integrates primary healthcare services with home and community-based care, offering an integrated approach to addressing the healthcare needs of older adults.

### 2.2. Study Population

Using a social constructivist perspective, this study seeks to explore participant experiences and their perceptions of the effectiveness and challenges encountered during the integration of health and elderly care services (yi-yang-jie-he) as part of a program. Focusing on older people’s care needs, the elderly care team in this case was initially formed around nurses. As the project developed, the family doctor project was introduced, and the team recruited a male general practitioner. However, due to personal income and workload factors, this doctor had already been transferred and had worked in the general outpatient clinic during the field research. Currently, in addition to administrative staff and social workers, the medical team consisted of one doctor, two rehabilitation therapists, and six nurses. In fact, the Elderly Care Department had relatively fewer personnel when compared with the other departments in the center, and integrated elderly care was not its focal point. As a community health care center, its main focus remained on primary medical care, outpatient and inpatient services, and public health services.

As a matter of fact, not only did the Elderly Care Department this case investigated faced difficulties in recruiting general practitioners, but the entire community healthcare center also struggled with a shortage of GPs. This aligns with issues such as inadequate training and promotion support for GPs in China’s primary medical system. Furthermore, the case study of this paper is a non-profit organization. Unlike government-supported public community healthcare centers, this institution needs to compete fiercely with its other counterparts to obtain government financial support. In the meantime, it also faces stricter governmental regulations, which has reduced the willingness of doctors to join primary care facilities and constrained the development of the institution. In order to gather comprehensive information, this study recruited all the members of the medical team as interview participants. Besides doctors and nurses, there were three management staff. Two social workers were also included, and they were responsible for running the day care centers (see Table 1).

### 2.3. Data Collection

Observational data

The data primarily stemmed from observations and semi-structured interviews conducted with selected staff members. All participants were informed of the field study and interview procedures, and they all provided their consent to participate. Observational data were gathered by researchers via the participant-as-observer approach [30]. One researcher fully engaged with the participants during their home visits, rounds, staff meetings, and other daily routines, becoming an active member of the team [31]. To mitigate the Hawthorne Effect, several strategies were implemented. The aforementioned researcher ensured absolute transparency with participants regarding the study’s objectives, and they debriefed them afterward regarding the field data. Observations were conducted in natural settings while the participants were engaged in actual work. Data from the initial month of the field study were disregarded to allow subjects to acclimate to being observed, with the subsequent observations then being utilized for data analysis. Observation periods were extended to six months to further minimize observer bias.

Semi-structured interviews

One objective of this study was to elucidate the decision-making processes at the hospital management level, with the aim of identifying the underlying integration issues and assessing the potential generalizability of this care model across similar units within the broader population. Qualitative assessments also encompassed the multidisciplinary cooperation and cross-level collaboration between the organization and the government, and these were aimed at revealing the evolution of community-based integrated care practices in China.

Semi-structured interviews were also conducted in this study, which combined a pre-determined set of open questions with the opportunity for the interviewer to explore the above themes or responses further. The interview guide was developed by drawing upon the literature from primary care, gerontology, community care, and social care [32]. Participants were interviewed by a researcher trained in qualitative interview techniques. They were encouraged to freely express their perspectives on the state of elderly care and community care in China. No prior relationship or familiarity existed between the interviewer and the participants. The interviews were conducted one-on-one wherever it is convenient for the participant, such as in their office. Each interview lasted approximately one to two hours, with around 1–2 sessions per participant. The research questions are shown in Table 2.

The interviews, conducted in Mandarin, lasted approximately 40–60 min each and were audio-recorded. Subsequently, the interviews were transcribed verbatim into Chinese text, coded, and then translated into English for further analysis. Thematic analysis was employed to identify the recurring themes within the data [33].

## 3. Results

Before performing thematic analysis, we identified three broad domains: policymaking, care model development, and barriers. A thematic framework outlining the key topics was then developed. As the data were structured, several themes and subthemes emerged. For each theme, a corresponding table was created (see Table 3), and the data were coded accordingly. The relationships among codes, subthemes, and themes were also described in detail.

Four key themes emerged from the data analysis. The first two themes centered around the service delivery model and development strategies within the community care scheme. The third theme highlighted the importance of the person-centered guiding principle, while the fourth theme illuminated the challenges and hardships encountered by the formal caregivers within the current community care system in China. Below are the details of each theme.

### 3.1. Theme 1: Policy-Driven and Resource Integration

The integration of elderly care work involves trans-disciplinary and cross-sectoral collaborations. According to government guidance, integrated elder care encompasses various components such as community stations and day care centers providing temporary care or nursing services, respite services offering relief to family caregivers, home-based medical care encompassing basic nursing care, rehabilitation exercises, and counseling, as well as home services like house cleaning and personal care [34]. These services are administered by different sectors, including hospitals, social work stations, disability resource centers, and elder care facilities. Each organization typically specializes in a particular area, and integration involves coordinating efforts among these diverse actors.

In practice, most institutional elderly care services in China are organized in three main ways: public social welfare facilities established and operated by the government, nursing homes owned by either the public or private sector, and other private residential care facilities [22]. However, these entities may lack recognized medical qualifications. Consequently, they must establish pathways for referrals with hospitals, rehabilitation centers, and hospice facilities when older individuals require medical attention. Additionally, they collaborate with social work stations and volunteers to provide social work counseling and other social or recreational activities.

Unlike the mere integration among disparate service providers, the HS CHC proactively embraces policies and integrates a diverse array of services into its own platform. For instance, in 2020, the Guangzhou government initiated an action plan for upgrading integrated elderly service centers (Yikang Center) in towns and cities. This plan seeks to coordinate community-based elderly service agencies and elder care institutions. At the end of 2020, the HS CHC successfully obtained an operating license from the civil affairs department and established a nursing home, a meal delivery service center, nine community day care centers, and several social work stations.

‘Not many medical institutions would provide elderly care services, though they may have a geriatric department. The social work stations in the community are the major actors to offer such services. We are the first community health center to take on specialized elderly care services, and this gives us more opportunities. At first, we saw that our patients and the whole neighborhood are aging. What these people need is different from general medical care. We started from practicing family doctor and home-based care. You can see we have necessary medical supports from the center, and this is what those social work stations do not have’.(Case 1)

Integrating resources from various government departments and securing sustainable financing are two pivotal strategies for its development. Program leaders collaborate closely with local government departments to garner support, such as the subdistrict office, local health commission, and civil affairs department. Additionally, the team actively engages in municipal and grassroots philanthropic activities to raise funds.

‘The center registered as a non-profit organization when it turned into an independent institution. Therefore, we are qualified to apply for philanthropy activity funds. Our applications are always around older people’s care needs, like chronic pain and pain control, family doctor, frail older people care, and hospice at home. We have been doing this for seven years, and made it a sustainable way for fundraising’.(Case 3)

With policy and financial support, the HS CHC has established informal caregiver career support and training, health education and promotion programs, and volunteer and community support initiatives. These efforts not only ensure integrated services in a continuum, but they also empower elderly residents and promote community participation.

### 3.2. Theme 2: Primary Care-Led and Internal Integration

As a primary care provider, the community health center occupies a unique position to identify and manage patient health statuses, fostering ongoing relationships with them over time. The HS CHC effectively coordinates its multidisciplinary resources to address the holistic needs of elderly patients, drawing upon its primary care services. These encompass a wide range of medical areas, including general practice, traditional Chinese medicine, gynecology, pediatrics, stomatology, ophthalmology, otorhinolaryngology, and physical examinations, among others. For instance, in 2020 alone, the HS CHC recorded approximately 320,000 general outpatient service attendances, with over 80,000 cases involving elderly patients. The center ensures a seamless transition throughout the care pathway, facilitating transfer services to secondary or tertiary hospitals when necessary.

‘The center has a well-developed primary care system, and it plays a significant role in public health to optimize the overall delivery of healthcare service in the neighborhood. This is a foundation for our elderly care services’.(Case 1)

‘Medically speaking, older people or people at the last stage of their lives are considered to have little value for treatments at some circumstances. Therefore, some hospitals or doctors are unwilling to take in such patients. But in our case, we can still help them even in their homes. We have experienced nurses in palliative care, and we built a palliative care ward. What we offer do meet the needs of some older people and their families’.(Case 6)

As the neighborhood population ages, there is a growing demand for long-term care services. Recognizing this need, the HS CHC established a Geriatrics Department in 2018, providing 220 beds for patients with chronic diseases and chronic wounds. Additionally, a Palliative Ward was established to support patients whose medical conditions do not respond to curative treatments, ensuring continuous care across the remainder of its inhabitants’ lifespans.

At the community level, patients are affiliated with primary care through their family doctor. The Family Doctor (FD) system, or Family Physician (FP) system, was launched in 2009, aiming to provide proactive, continuous, and comprehensive health management by establishing service contracts with residents [35]. Family doctor services at the HS CHC primarily focus on managing common chronic diseases such as cardiovascular disease, hypertension, hyperlipidemia, and chronic pain. These services encompass diagnosis and treatment in family medicine, nurse clinics, pharmacy advice on medications and consultations, as well as evaluations for muscle pain. In addition to outpatient and inpatient care, elderly patients can receive regular treatments at home. This approach makes it more accessible for them to engage with health service providers for various needs, including evaluation, the management of personal health records, dietetic assessments, nutrition counseling, and monitoring. These functions are managed by the Elderly Care Department (the yi-yang-jie-he unit).

Furthermore, the HS CHC cultivates a robust primary care workforce, comprising allied health professionals, nurses, social workers, and informal caregivers. In the practice of integrated care, the Case Management Programs (CMP) strategy is employed to address the needs of frequent users of healthcare services with complex requirements [36]. A doctor or physiatrist, alongside a nurse, is designated as the case manager to assess, plan, implement, coordinate, and prioritize services according to individual needs, social workers, and informal caregivers [37]. To enhance coordination further, the HS CHC has established social work stations under the Elderly Care Department (the yi-yang-jie-he unit). Informal caregivers receive training from the case management team and function as community partners, contributing to multidisciplinary collaboration and quality assurance within the workforce.

‘Our understanding of “integration” has undergone a process of transformation. Initially, we established various teams to meet the multiple care needs of elderly people. However, gradually we realized that this was only the most basic form of integration, and we faced the challenge of truly connecting the resources within the center. For example, after patients are discharged from the hospital and return home, their care needs are assessed by our care team, and then passed to the family doctor team to continually provide elderly care bed services, home-based continuing care, and long-term care insurance service in the patients’ own homes. At the public health level, elderly individuals with identified needs during physical examinations are transferred to the chronic disease management team, further assessing whether they require outpatient or inpatient services. In this way, we have completed a closed-loop service from outpatient to inpatient care to home care, both at the community elderly care and the institutional medical care level’.(Case 2)

The practice of the HS CHC indicates that the significance of integrated care lies not only in consolidating fragmented medical and social service resources, but also in meeting the personalized care needs of each individual throughout their lifecycle within the primary healthcare setting. Integrated care is more about the integration of person-centered needs in this case.

### 3.3. Theme 3: A Shift from Disease-Centered to Person-Centered Care

The HS CHC model exemplifies the reorientation of the health system toward a “community of persons”, thereby repositioning the elderly within a lifelong health journey. Person-centered care is embraced as an approach that recognizes individuals as holistic beings with diverse needs and goals in their care [38]. Transitioning from a disease-centered to a person-centered approach, the HS CHC establishes a framework that addresses the various contributing factors to an age-friendly community. This framework places a strong emphasis on understanding the unique circumstances of individual elders within the context of their life experiences, family dynamics, social connections, and the community in which they reside.

The district where the HS CHC is located is heavily industrialized, housing numerous dockyards, petroleum refineries, foundries, and power plants. In the 1990s, the HS CHC served as the staff hospital of a state-owned shipyard enterprise. The neighboring community was structured, integrated, and governed through the workplace, exhibiting the typical characteristics of a *danwei* system [39]. Particularly for residents over 70 years old, their life trajectories and social identities were shaped by the socialist workplace and its associated activities. Many residents in the A Subdistrict were formerly coworkers and old acquaintances, fostering a sense of familiarity and community.

With a comprehensive understanding of people’s circumstances, the HS CHC integrates social networking into its community-based care model. Recognizing the traditional role of older people receiving care within intergenerational families, the center recruits and employs capable members as part-time caregivers. This approach allows individuals to care for their own families while also assisting nearby neighbors. As a result, medical service providers and community members collaborate as co-producers of care. The center establishes pathways that foster partnerships, engaging and empowering people within the community.

‘As family nurse practitioners, most of our daily work is to visit our home-based patients. We check their health status, maintain patient records, perform physical exams, etc. We also perform some basic treatments, like minor injuries, infections, or urine catheter change. Our work mainly is about preventive care as well as treatment of various health conditions’.(Case 7)

‘We give the part-time caregivers training and supervise their daily care work. With each patient, a care plan is made. The part-time caregivers are asked to follow the plan, and we take notes on their care, which is activated during family visits. Besides personal and medical care trainings, the part-time caregivers also need behavioral and mental health care trainings. Especially for the medical part, not many informal caregivers are properly trained. A common problem we find is that for some frail older adults, they often suffer from pressure sore. In fact, pressure sore problems can be avoided if they can receive care with good quality such as, for example, getting turned and positioned correctly. It’s quite simple to proceed, but unfortunately, some informal caregivers are just unaware of that’.(Case 4)

Achieving the goal of whole-person health requires not only medical resources, but also additional health and community resources. The care team has observed that many elderly individuals with chronic illnesses may experience partial or total impairment of their bodies. As a result, day training, residential, and community support services are provided to assist elderly individuals with disabilities in enhancing their physical, mental, and social capabilities. Additionally, the team collaborates closely with the local Disabled Persons’ Federation to help elders apply for special allowances. Chronic pain is a prevalent symptom among aging populations. To address this issue, the team developed a long-term educational program aimed at educating elders and informal caregivers about pain management, interventions, and self-help measures.

### 3.4. Theme 4: Barriers and Challenges for Integrated Community Care

One significant challenge inherent in community care settings is the tendency for service development and change. For institutions, securing sustainable financial support is crucial. Program management endeavors to make the program financially viable and sustainable for long-term development. However, implementing home-based and community care presents additional challenges in maintaining program viability and profitability. Apart from the costs associated with human resources, the scarcity of medical resources and the comparatively higher service prices render home-based and community care financially more demanding than institutional care. Additionally, restrictions on the use of medical equipment in community settings objectively constrain the availability of medical services that can be delivered at home.

In this case, financial resources primarily originate from three sources. The primary support stems from the government’s procurement of elderly care services, such as the Yikang Center project. Funding for this initiative is allocated by the city civil affairs department and overseen through its hierarchical bureaucratic structure. The second source comprises governmental subsidies for each case of community home care beds and family doctor beds, and these are disbursed and monitored by the local health commission and civil affairs department, respectively. These two funding streams sustain the institution’s day-to-day operations. Additionally, the HS CHC engages in venture philanthropy endeavors and solicits charitable donations as supplementary fundraising avenues.

In managing change, it is imperative to involve all staff in the mindset shift process and to allocate sufficient time to integrating new practices effectively.

‘Though the program has been running for several years, some colleagues from other departments still do not fully understand the idea and significance behind it. For some doctors, they are used to the way that patients would come to them for help. While working in the community environment, we as medical professionals have to knock on people’s doors and ask if they need help. It’s not easy for some doctors to adjust their mindset. Therefore, we see obstacles within the organization. For example, when the managements sit together and make the next year’s budget, we have to argue with some colleagues to get a bigger share. Afterall, we have not made the program fully independent and self-sustainable yet’.(Case 6)

Managing the lives of the elderly entails navigating multifaceted networks and engaging with various professional groups. However, the integration of health and care services has been hindered by fragmentation. Sectors involved in integrated care are overseen by different government departments, such as the subdistrict office, local health commission, and civil affairs department. As a result, standards, regulations, and policies are established and supervised by disparate entities. Achieving integration also entails navigating complex coordination among policies, organizational structures, financial models, cultures, and legal responsibilities, presenting yet another challenge.

‘Policies are made by different government departments. Money comes from different sources. Nearly every report from the supervising departments is a different one, and we have to spend a lot to time on paper work. At the policy making and supervision level, better integration is needed’.(Case 8)

For frontline care workers, delivering integrated and patient-centered care necessitates treating all patients, irrespective of age, as individuals with unique strengths, histories, and needs. Establishing outcome measures that effectively address the diverse behavioral health and primary care needs of older populations remains a challenge. The National Health Commission of China has issued comprehensive guidelines for integrated care. However, in practice, decision makers face difficulties in planning combined home health and social services.

‘As a social worker, I have to consider a number of working regulations and user-related necessities. I also need to synchronize all types of services. My daily work includes housekeeping, laundry, and bath assistance. Some older residents cannot figure out how to use their smart phones, and they will come to us for help. And if they have to visit some places, we also keep them company. Therefore, there are a lot of different tasks, and care routing and scheduling is challenging. Most of the time I’m on my own, and the work is heavy’.(Case 9)

## 4. Discussion

The findings of this study highlight that the integrated care model and development strategies of the HS CHC are centered around its primary care services. Figure 1 provides a visual depiction of the model. Unlike the disease-focused, hospital-centric approach, its person-centered care delivery enhances integration between providers and recipients. A care pathway tailored for older populations restructures services around individuals and offers care at all stages of their journey, from healthy, active aging to frailty and end-of-life stages. Despite challenges surrounding the integration of various policies and multidisciplinary work groups, the service delivery model has the potential to bridge these gaps and provide older adults in the community with integrated care to address their needs.

Elderly individuals account for a significant portion of healthcare service utilization, yet the current healthcare system inadequately addresses their needs. Previous studies have demonstrated that effective primary healthcare not only provides continuous care for chronic conditions, but that it also involves a diverse array of healthcare providers in delivering comprehensive care to patients [40]. Despite primary healthcare being considered an ideal setting for older adults to receive coordinated care, the delivery of healthcare has become fragmented in many countries due to structural and funding-based barriers between primary and community care [41,42]. China faces similar challenges, including a fragmented health and care governance structure [43]. This case study yields similar findings, highlighting the presence of these barriers.

However, the integration model revealed in this case study suggests that, within the primary and community care setting, the community health center itself can take a leading role in integrating various sectors. Since China initiated the integration of health and care policies in 2016, there have been 90 national pilot sites showcasing different integration models [44,45]. Institutional-level care integration includes cooperation between health and care institutions, the integration of healthcare into elder care facilities, and the inclusion of social care within healthcare facilities [46]. For instance, the Luohu District Hospital Group in Shenzhen City, Guangdong Province, integrates five hospitals (including one geriatric hospital) and 23 community health centers in the district, facilitating vertical integration between acute and primary care [47]. Unlike the Luohu model, which is a large-scale medical industry conglomerate model with direct government support, this case demonstrates the potential for integration within a single primary care facility. With community medical care as its foundation, supported by home-based care and complemented with institutional care, this model encourages self-care for most elderly patients in the community with stable chronic conditions or mild episodic illnesses.

At the heart of the community care concept lies the principle of enabling individuals in need to remain in their own homes and stay connected to their communities, which is often referred to as “aging in place” [48]. The implementation of a community-based approach and the shift from a disease-centered to a person-centered care (PCC) model offer significant advantages for aged care. PCC is widely acknowledged as a standard for quality care as it is particularly beneficial for older individuals who often have multiple care needs and complex health conditions [49]. The integrated community care model at the HS CHC focuses on addressing the diverse needs of older adults by combining medical, social, and community-based services. Seniors receive more personalized care, including regular check-ups, preventive care, chronic disease management, and specialized treatments. The integration of the medical and social care workforce has proven to be effective within community settings, providing both formal and informal caregivers with a deeper understanding of each individual case. At the community level, care integration primarily advocates for the contracted services of family physicians. According to governmental guidelines, residents can sign contracts with “family physician/general practitioner-led primary care teams” in community health institutions, which are mostly community health centers run by local governments. These teams offer basic care services, including medical care, preventive care, and health management, such as health assessment, rehabilitation guidance, and home-based medical care [50].

In addition to providing comprehensive health services, the primary care team also addresses residents’ well-being and their other home and community-based needs. This study highlights the various opportunities and challenges encountered during the implementation of integrated community care. It emphasizes the crucial role of establishing and nurturing multidisciplinary teams and fostering collaboration to effectively overcome obstacles. In this context, the case manager emerges as a central figure: one who is responsible for developing personalized care plans and coordinating the efforts of the care team. Furthermore, family doctors and nurses play critical roles in addressing the daily care requirements of older patients, serving as vital connectors between home-based care and primary care, thus ensuring seamless continuity and comprehensive support for the elderly population.

## 5. Limitations

This study has several limitations. Firstly, the small sample size of purposefully selected participants limits the generalizability of the findings. These participants are practitioners of integrated community care services and may hold stronger opinions, potentially introducing voluntary response bias. Additionally, this study did not assess the health impact of the integrated community care service on older adult users, which may have restricted our understanding of their experiences and perceived benefits from the provided services. Furthermore, it is important to note that this study focused on a single integrated community care model, and there are additional care models that warrant further investigation and study. Lastly, there is a need for further quantitative research to generate empirical evidence on whether this service model effectively addresses the existing social inequalities in community care needs in China.

## 6. Conclusions

This study meticulously examines the integration of an integrated community care system within a local community in China. The integration process is scrutinized in detail, shedding light on the challenges and facilitators involved. The HS CHC integrated community care service model stands out as a notable example, showcasing the pivotal role of primary care as a cornerstone in successfully implementing elderly care within a community setting. By incorporating essential components such as family doctors and social work, a primary care facility can effectively integrate fragmented resources and leverage government support to its fullest extent. Within China’s top-down political–social structure, the government takes a leading role in advocating for and advancing medical and social welfare affairs. However, non-governmental organizations can also make significant contributions by actively engaging in and contributing to public healthcare and social care services, particularly for older populations. Therefore, the adoption of a private organization-led approach to medico-social integration care within the community holds immense potential as a service delivery model for effectively addressing a wide array of elderly care issues.

The aging population poses similar challenges for countries worldwide, particularly concerning health, long-term care, and welfare systems. Many policymakers advocate for prolonged residence in private homes as people age, prioritizing community care over relocation to long-term care facilities. This shift underscores the growing importance of community-based care for the elderly. European countries, for example, have significant room for improvement in organizing and coordinating long-term care services [51]. Lessons learned from this case study can be applied elsewhere, including the adoption of common quality standards and integration with primary healthcare systems. In contrast to China, where elder care is typically centralized within multigenerational family households, over 30 percent of European elders live alone [52]. Consequently, the responsibility for elder care, both financial and labor-related, falls more heavily on the state rather than individual families. Given the need for highly integrated and cost-effective community elderly care, relying solely on the market or civil society may prove ineffective. Learning from Chinese practices, which emphasize comprehensive regulation to integrating fragmented sectors and resources, could be valuable both at the national and municipal levels. Moreover, additional measures are required to ensure universal access to services, particularly by making care more affordable for low-income families to adequately support their elderly relatives.

## Figures and Tables

**Figure 1 ijerph-21-00680-f001:**
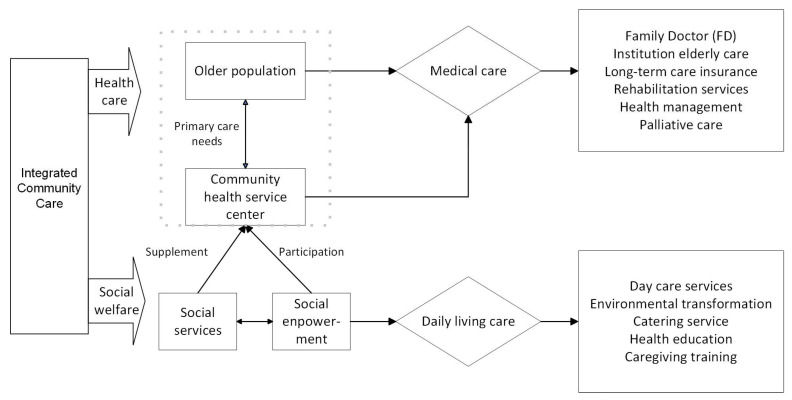
Care model of the HS CHC.

**Table 1 ijerph-21-00680-t001:** Characteristics of the semi-structured interview participants.

Case Number	Gender	Age	Role and Profession
1	Female	72	Administrative staff
2	Male	68	Administrative staff
3	Female	38	Administrative staff
4	Male	30	Family doctor
5	Female	23	Physician
6	Female	36	Physician
7	Female	49	Nurse
8	Female	23	Nurse
9	Female	29	Nurse
10	Female	32	Nurse
11	Female	30	Nurse
12	Female	41	Nurse
13	Female	41	Social worker
14	Female	54	Social worker

**Table 2 ijerph-21-00680-t002:** The semi-structured interview questions.

First Interview: The purpose of this interview is to have the interviewee briefly describe their background and their roles and responsibilities in community-based integrated health and eldercare services.	
1	Please briefly describe your background, such as age, educational background, past experiences, etc.
2	Could you share your work experience, including motivations for engaging in medical/social services/integrated health and eldercare/elderly care, years of experience, key insights gained from work, etc.?
3	What have you experienced since joining the community-based integrated health and eldercare project?
4	What are your daily work routines?
5	Have you experienced collaboration with colleagues from other fields within the project? What kind of collaboration is mainly involved?
6	Have you encountered any challenges/difficulties in your daily work? How did you overcome them?
7	What special skills/techniques/methods do you think are needed for carrying out elderly integrated health and eldercare services in the community?
Second Interview: The purpose of this interview is to understand the interviewee’s perspectives on the development and actual effects of community-based integrated health and eldercare services.	
1	Have you ever worked in elderly care at institutions such as hospitals/nursing homes/daycare centers? What do you think are the differences between institutional care and community care? What are the respective advantages/disadvantages?
2	What do you think about the current actual effects of community-based integrated health and eldercare work? Does it meet the care needs of the elderly?
3	What problems/obstacles do you think currently exist in community-based integrated health and eldercare work? How should they be addressed?
4	In what areas do you think community-based integrated health and eldercare currently needs improvement/perfection?
5	How do you think the country/government should assist in improving community-based integrated health and eldercare work?
6	Based on your practical work experience, could you discuss the specific caregiving needs of the elderly in the community? Which of these needs require improvements?

**Table 3 ijerph-21-00680-t003:** Coding for the focus areas, themes, and subthemes.

Focus	Themes	Subthemes
Policymaking	Health governance	Governance levers and structures
		Government guidance and surveillance Purchasing services
		Public–private partnership Medical–social collaboration model
Care model development	Health and social systems	Primary and hospital services
		Personal and population health services Long-term and community care Social services Care settings Community participation
	Care workforce building	Organization and member information
		Multidisciplinary collaboration
		Service providers and healthcare professionals Connections Care pathways
	Providers and source of care	Formal and informal caregivers
		Family Doctor (FD) Assessment of care needs Quality assurance
Barriers	Challenges and hurdles	Financial means
		Budget arrangements
		Member recruitment
		Payment and career development
		Work intensity and pressure

## Data Availability

The data presented in this study are available on request from the corresponding authors. The data are not publicly available due to privacy reasons.

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
