# Peer review of "Delivering Integrated Community Care for the Elderly: A Qualitative Case Study in Southern China"

_ijerph, 2024, doi:10.3390/ijerph21060680_

Round 1

Reviewer 1 Report

Comments and Suggestions for Authors

The article deals with the very topical and interesting issue of integrated community care. It is of considerable interest in China, where the problem will be particularly intense. The methodology of the study is correct. The conclusions and recommendations are well formulated. Minor substantive reservations (1) Sampling - the authors are aware of this weakness and point it out in the text. Particularly important is the fact that the sample included two doctors with relatively little professional experience (aged 30 and 23 years); (2) introduction to the topic - the article would have been enriched by a broader presentation of the problem of ageing and the care system in China, with figures, whether the on-site care is nationwide or a pilot project, whether any standards or quality measures have been defined; (3) the authors refer to the need for cost-effective use of resources, but do not discuss this issue further. Integrated community care is financially more expensive than institutional care. However, the authors do not indicate how the observed institutions deal with this issue.

Reviewer 2 Report

Comments and Suggestions for Authors

1.     B community Health Center, A subdistrict…, both terms need to be explained more in detail to facilitate the readers to understand (lines88-89).

2.     What are the differences between Integrated care model and holistic approach? (lines 76-78, 98-99)

3.     The reasons or criteria to recruit the participants need to be described in the study population.

4.     Observational data were collected by shadowing .., how to conduct these procedures? These participants knew to be recruited in the study, how to prevent the Hawthorne effect?

5.     How many questions of the interview?

6.     The functions of BCHC seems very powerful. What kind of unit for it? (lines 217-220, 233-242)

7.     Did the participants in lines 221-232 be hospital at home?

8.     Finally, the authors want to express what is the true meaning of “integration”? (lines 244-254)

9.     What are the connections among lines 265-292?

10. The term caregivers are too broad, including professionals, semi - professionals, and non- professionals. The readers are easy to be confused.

11. The term bedsore has to be changed to pressure sore.

12. Please explain the meanings between comprehensive care and holistic approach. (lines 99, 353)

Reviewer 3 Report

Comments and Suggestions for Authors The introduction must contain more information about medical personel, not only abouts adulst / old person in China. You should show bacground and add information about healthcare not only about patients.   Material and methods: In the materials and methods section, you describe the population of individuals over the age of 60, but you provide insufficient information regarding the population of medical personnel with whom you conducted interviews. Please supplement the sample selection, describe the questionnaire, research methodology (interview), types of questions whether they were open-ended or closed-ended questions, as this is not explained. Why were there only 9 interviews? Please describe the difficulties in recruiting respondents, on what basis they were chosen, why so few, and how the interviews were conducted. In the methodology, it is not described exactly what you did with the interviews. Of course, with such a small sample, there is no possibility of statistical inference. However, I believe that the interviews should be complemented by quantitative analysis conducted on a much larger group.   Results: The results are well-described, divided into sections, which provides clarity and transparency.     The discussion lacks references to similar results, for example, from Europe. Please supplement. The conclusions are okay.  

Please provide the ethical committee approval number, or if it is not required, an explanation as to why.

Round 2

Reviewer 3 Report

Comments and Suggestions for Authors

In the first review, I pointed out that the results are only described, they are too extensive, and have not been prepared in graphical form. I understand that this is a quantitative study, which is difficult to display graphically, but nevertheless, I suggest improving the results.
